# Bergamot Polyphenolic Fraction for the Control of Flupyradifurone-Induced Poisoning in Honeybees

**DOI:** 10.3390/ani14040608

**Published:** 2024-02-13

**Authors:** Roberto Bava, Carmine Lupia, Fabio Castagna, Stefano Ruga, Saverio Nucera, Rosamaria Caminiti, Rosa Maria Bulotta, Clara Naccari, Cristina Carresi, Vincenzo Musolino, Giancarlo Statti, Domenico Britti, Vincenzo Mollace, Ernesto Palma

**Affiliations:** 1Department of Health Sciences, University of Catanzaro Magna Græcia, 88100 Catanzaro, Italy; roberto.bava@unicz.it (R.B.); studiolupiacarmine@libero.it (C.L.); rugast1@gmail.com (S.R.); saverio.nucera@hotmail.it (S.N.); rosamariacaminiti4@gmail.com (R.C.); rosamaria.bulotta@gmail.com (R.M.B.); c.naccari@unicz.it (C.N.); carresi@unicz.it (C.C.); v.musolino@unicz.it (V.M.); britti@unicz.it (D.B.); mollace@unicz.it (V.M.); palma@unicz.it (E.P.); 2Mediterranean Ethnobotanical Conservatory, 88054 Sersale, Italy; 3Department of Pharmacy, Health and Nutritional Sciences, University of Calabria, 87036 Rende, Italy; giancarlo.statti@unical.it; 4Center for Pharmacological Research, Food Safety, High Tech and Health (IRC-FSH), University of Catanzaro Magna Græcia, 88100 Catanzaro, Italy

**Keywords:** honeybee (*Apis mellifera*), flupyradifurone, poisoning, bergamot (*Citrus bergamia* Risso et Poiteau) bergamot polyphenolic fraction (BPF), natural products, animal health and welfare

## Abstract

**Simple Summary:**

Pesticide poisoning is a particular concern for honeybee farms. In addition to causing mortality due to acute, subacute, and chronic phenomena, poisonings debilitate honeybees by making them more susceptible to damage by other pathogens and environmental stressors. For this reason, it is important to seek solutions that can mitigate the effects of these phenomena. In this laboratory study, it was shown how dietary supplementation of the bergamot polyphenolic fraction is able to mitigate the toxic effects of flupyradifurone, also increasing the survival probability of honeybees.

**Abstract:**

Flupyradifurone (FLU) is a butenolide insecticide that has come onto the market relatively recently. It is used in agriculture to control aphids, psyllids, and whiteflies. Toxicity studies have decreed its low toxicity to honeybees. However, recent research has challenged these claims; oral exposure to the pesticide can lead to behavioral abnormalities and in the worst cases, lethal phenomena. Compounds with antioxidant activity, such as flavonoids and polyphenols, have been shown to protect against the toxic effects of pesticides. The aim of this research was to evaluate the possible protective effect of the bergamot polyphenolic fraction (BPF) against behavioral abnormalities and lethality induced by toxic doses of FLU orally administered to honeybees under laboratory conditions. Honeybees were assigned to experimental groups in which two toxic doses of FLU, 50 mg/L and 100 mg/L were administered. In other replicates, three doses (1, 2 and 5 mg/kg) of the bergamot polyphenolic fraction (BPF) were added to the above toxic doses. In the experimental groups intoxicated with FLU at the highest dose tested, all caged subjects (20 individuals) died within the second day of administration. The survival probability of the groups to which the BPF was added was compared to that of the groups to which only the toxic doses of FLU were administered. The mortality rate in the BPF groups was statistically lower (*p* < 0.05) than in the intoxicated groups; in addition, a lower percentage of individuals exhibited behavioral abnormalities. According to this research, the ingestion of the BPF attenuates the harmful effects of FLU. Further studies are needed before proposing BPF incorporation into the honeybees’ diet, but there already seem to be beneficial effects associated with its intake.

## 1. Introduction

Honeybees comprise several subspecies bred for the production of valuable food products. These insects also play an important ecosystem role through pollination activity [1]. The increasing use of pesticides can affect pollination and honeybee activities in agricultural and natural habitats. Pesticide misuse is a major contributor to the decline in honeybee populations and insect biodiversity [2,3]. It is therefore important not to underestimate the consequences of pesticide use. It is now common knowledge that the application of pesticides must be limited and regulated to ensure the well-being of pollinators [4,5].

Progress in this direction has been made by banning the use of some particularly dangerous pesticides, such as neonicotinoids and dichloropropene [6,7]. Furthermore, several pesticides are currently less effective because the pests they are supposed to control are becoming resistant [8]. As a result, new generations of pesticides are making their way onto the market. The new pesticide products are of interest as they are characterized by a low hazard quotient (HQ) for honeybees. This quotient measures the overall risks associated with the quantities of pesticides in the environment or in the hive matrices, whether real or expected. The pesticide LD50 values are then connected to HQ levels [9]. Among the new pesticides introduced relatively recently is flupyradifurone (FLU).

This pesticide, an active component of the Sivanto Prime^®^ product, is used to manage a variety of pests, such as aphids, psyllids, and whiteflies. Similarly to neonicotinoids, FLU interacts with the nicotinic acetylcholine receptor (AchR) in the nervous system of honeybees [10]. FLU is a butenolide insecticide that has a good safety profile for honeybee colonies, according to 2014 USEPA data [11,12]. The United States’s Environmental Protection Agency (US EPA) claims that FLU is non-volatile in water, moist soils, and dry surfaces, so there is no issue with the diffusion determined by the wind [11]. In addition, available environmental fate data indicate that FLU tends to dissolve quickly from the point of application. However, its ecotoxicological assessment is questionable.

Recent studies have discovered the existence of FLU in environmental matrices to which honeybees are exposed, such as pollen and flower nectar; these data support its persistence in the environment. Acute contact exposure to FLU is not dangerous to adult honeybees, but FLU is very harmful to honeybees when administered orally [11]. After consuming residues even at fairly low exposure levels, 50% of honeybees will experience acute death [11]. Exposure to FLU in the field for an extended period of time may also harm honeybee behavior and survival [13]. It was shown by Hesselbach and Scheiner in 2019 that honeybee taste and cognition were compromised by acute exposure to a high, non-field-realistic FLU dosage (1.2 µg bee^−1^) [14]. However, since it is considered safe for honeybees, FLU can be applied to crops that are in bloom even while honeybees are present [15]. A worldwide study team has assessed the long-term impacts of the new insecticide FLU in order to better define these elements.

The researchers used a novel risk assessment method that improves the protection of the pollinating population and decreases the underestimate of pesticides’ long-term risks. The experiment aimed to determine the behavioral and lethal effects of exposing several honeybee subspecies to different concentrations of FLU. For the research, seven laboratories from six different countries in Europe and North America have been enrolled. The study’s findings show how extended exposure to this substance can increase honeybee mortality. Even at low doses of FLU (101 times lower than those found in previous shorter-term experiments), it could have a negative impact on honeybee survival and behavior. At the examined doses, this pesticide increases the percentage of honeybees that display abnormal behavior, such as confusion and hyperactivity. The study shows that there are detrimental effects on honeybee survival and behavior (such as poor coordination, hyperactivity, and tiredness) even at real concentrations of FLU in the field.

Foragers proved to be the more vulnerable to this pesticide (4-fold effect) when compared with the other honeybees of the hive; all types of workers are substantially more sensitive to the FLU in summer than in spring [13]. This described study raises awareness of the inadequacy of risk assessment protocols and pushes for the urgent need to develop comparative protocols for different hive bee castes as well as for different pollinating insect species. As things stand at present, in order to immediately protect honeybees, it is imperative to devise practical methods to strengthen the body’s response to chemical poisoning. Natural products (NPs) and their derivatives have been successfully used for the treatment of numerous diseases and show great promise as demonstrated by numerous research studies in veterinary medicine [16,17].

The bergamot plant (*Citrus bergamia*, Rissu) is native to Southern Italy’s Calabrian area. The bergamot (*Citrus bergamia*, Rissu) peel’s essential oils are highly defined and used in a wide range of products from food, medicinal, and cosmetic industries [1,11]. According to previous research, the volatile phytochemical content of the essential oil can range from 66 to 93%. These phytochemicals include linalool (2–20%), linalyl acetate (15–40%), and monoterpenes (25–53%). Waxes, pigments, coumarins, and psoralens are examples of non-volatile substances [18,19]. The bergamot polyphenol fraction (BPF), a polyphenol-rich fraction with antioxidant, anti-inflammatory, lipid-lowering, and hypoglycemic properties, is produced from the juice and albedo of bergamot (*Citrus bergamia*, Rissu) [20]. Juice and albedo contain a unique combination of flavonoids and flavonoid glycosides, including neoeriocitrin, neohesperidin, naringin, rutin, neodesmin, rhoifolin, and poncirin. Due to these traits, the BPF seems to have a positive role in several pathophysiological pathways [21]. A recently published study showed that the BPF is able to mitigate the effects of deltamethrin intoxication [22]. In this study, the protective properties of the BPF against FLU-induced acute oral intoxication in caged honeybees were investigated.

## 2. Materials and Methods

### 2.1. Preparation of the Experimental Groups

This study was conducted at the Interregional Research Center for Food Safety & Health (IRC-FSH), Department of Health Science, University “Magna Græcia” of Catanzaro (Italy). As recommended by the guidelines, the experiment was conducted in summer to avoid having honeybees with altered physiology, which happens in early spring and late autumn [23].

Three *Apis mellifera ligustica* hives were the source of the experimental honeybees. Conventional beekeeping practices were employed to manage the colonies. Before harvesting the honeybees, standard inspection techniques were performed to ensure the health of the colonies. The honeybee harvesting procedure predicted the withdrawal of a brood comb of hatching honeybees. The comb was moved to an incubator (35 °C and 65–80% relative humidity) and emerging honeybees were collected from it after 12 h. This method allowed us to acquire honeybees of the same age, as established by standard protocols for toxicological study in honeybees [23].

The honeybees were mixed after brushing the combs, divided into groups of 20 and moved into cages. Routine techniques without anesthesia were employed to put the honeybees into the cages [24]. The experimental groups were maintained in the dark at 33 ± 2 °C and 70% RH. For treatment application, the cages were equipped with feeders inserted horizontally on the bottom of the cage.

The feeders were graded 2.5 mL sterile syringes for single use with capped ends. The 1-day-old emerging honeybees were housed in the cages and given a sucrose solution (50% of sugar in distilled water) from Day −1 to Day 0 before the beginning of the experiment. This phase allowed them to become used to the test environment. Subsequently, honeybees were fed with FLU and BPF alone, both mixed into sucrose solution, and BPF combined with FLU into sucrose solution. Honeybees received two oral doses of FLU. The sucrose solution was mixed with FLU dosages at the two different concentrations of 50 mg/L and 100 mg/L. As verified by Gao et al., 2023 [25], the concentration of 50 mg/L is sub-lethal, while that of 100 mg/mL is close to the median lethal concentration (LC50). In the combination FLU and BPF treatment, each BPF dosage (1 mg/kg, 2 mg/kg and 5 mg/kg) was administered in conjunction with each dose of FLU. Ten replicates were created for each group; accordingly, ten control groups (doses of 0 mg/mL honeybee) were also composed. Food consumption and survival rates were tallied every 24 h for the entire duration of the experiment (72 h). Furthermore, the frequency of abnormal behavior in the honeybees (1–4, 24, 48, and 72 h after treatment) was observed in relation to the treatments. To recapitulate, twelve experimental groups were established, as shown in Table 1 below.

### 2.2. Preparation of BPF

The southern Italian province of Reggio Calabria has a large bergamot (*Citrus bergamia*, Risso) cultivation industry. Located on a coastal strip up to 12 km wide from the shore, the majority of the bergamot (*Citrus bergamia*, Rissu) growing area is situated between Scilla to the west and Monasterace to the east [26]. In the area encircling the municipality of Bianco (Reggio Calabria, Italy), bergamot (*Citrus bergamia*, Rissu) fruits were collected for this study. Bergamot (*Citrus bergamia*, Rissu) juice was extracted from peeled citrus by squeezing. The juice underwent oil fraction depletion by stripping, purification by ultra-filtration, and loading onto a suitable polystyrene resin column able to absorb compounds related to polyphenols with molecular weights between 300 and 600 Da.

The polyphenol fractions were eluted using KOH solution. The basic eluate was incubated on a rocking platform to reduce the quantity of furocumarin. The time of the shaking was changed in accordance with the amount of furocumarin impurities. Following the extraction of the furocumarins, the phytocomplex that was left behind was neutralized by filtering on cationic resin at an acidic pH. It was vacuum dried (water content <10%), then cut to the right particle size (pass 70 mesh) to produce BPF powder. BPF powder was examined for flavonoids, furocoumarins, and other polyphenol contents. High-resolution mass spectrometry (Orbitrap spectrometer) was used to identify the flavonoid profile. Furthermore, all toxicological tests were conducted, including those for the presence of pesticides, phthalates, and sinephrine, which revealed the absence of known harmful compounds at detectable levels.

### 2.3. BPF Analysis

A 320 ppm solution was created by dissolving 1.1 mg of BPF powder in 1 mL of water and diluting it 1/10 (*v*/*v*). Following filtering, an autosampler was used to inject 1 µL of this solution into a UHPLC/HRMS Orbitrap Q-exactive mass spectrometer. Thermo Scientific (Rodano, MI, Italy) Dionex ultimate 3000 RS was used for chromatography. It was directly injected onto a Thermo Scientific Hypersil Gold C18 column (150 × 2.1 mm, 1.9 µm particle size), which was equilibrated in 95% solvent A (0.1% aqueous solution of formic acid), and 5% solvent B (methanol). The autosampler and column temperatures were kept at 20 °C and 24 °C, respectively. The solvent B concentration was linearly increased from 5 to 100% in 45 min, then from 45 min to 47 isocratic at 100%, returning to 5% at the end of 55 min and re-equilibrated in 5 min to achieve the 200 µL/min elution flow rate.

The run took 60 min in total, which included column washing and equilibration. For complete scan analysis (mass range 550–850 amu), electrospray with negative polarities at 35,000 resolving power (defined as FWHM at *m*/*z* 200), IT 100 ms, and ACG target = 500,000 was used on a Thermo Scientific Q-ExactiveTM mass spectrometer (Rodano, MI, Italy). Operating at resolution 17,500, IT 200 ms, ACG target 10,000, quadrupole isolation windows at 0.4 *m*/*z*, and collision energy fixed at 25, the data-dependent scan with an inclusion list (ddMS2) was conducted. The following were the source conditions: 2.9 kV spray voltage, 30 arbitrary units of sheat gas, 10 auxiliary gas, 280 °C for the probe heater temperature, 320 °C for the capillary, and 50 RF levels for S-Lns. Before starting the analysis, the device was calibrated using Thermo calibration solutions.

### 2.4. Concentration and Doses of FLU and BPF Treatments

The pesticide testing solutions were given to the honeybees in the form of contaminated sucrose solution. Deionized water was used to make stock solutions of FLU and BPF, which were then maintained at 4 °C. The final treatment solutions were made by blending stock solutions, FLU, and BPF with 50% (*w*/*v*) water sugar solutions. The dosages for the BPF therapy were 1 mg/kg, 2 mg/kg and 5 mg/kg. FLU treatments were made at 50 mg/L and 100 mg/L concentrations. BPF was added to the above-mentioned FLU concentrations to confirm the protective properties of the polyphenolic extract. These dilutions were prepared, securely wrapped in aluminum foil to protect them from light degradation and stored at a temperature of 6 to 2 °C. New treatment solutions were prepared at least once every three days. In the feeding solutions, there was never any precipitation. During the exposure period, all the feeders and their contents were replaced every 24 h to guarantee that the honeybees had adequate food.

### 2.5. Food Consumption and Abnormal Behaviors

The average daily intake of solution was determined. This estimation was carried out using feeders provided of scale. The daily ingested solution from each cage was adjusted for the number of alive honeybees in each cage. Since all experimental variables were retained between treatment groups—aside from the variable of interest—and systematic conservative errors were created between cages, according to the “Standard methods for maintaining adult *Apis mellifera* in cages under in vitro laboratory conditions”, it was not necessary to take into account the evaporation [24].

Behavioral disorders were classified and quantified in accordance with the official ecotoxicological guidelines established by the OECD expert group on ecotoxicology [27]. The proportion of honeybees exhibiting abnormal behaviors throughout time (1, 2, 4, 24, 48, and 72 h after treatment) and the number of mis-behaving honeybees per cage were assessed depending on the pesticide dose. A curved-down abdomen, hyperactivity, apathy, motion coordination issues, and moribund ness were taken into account. The cage served as the replication unit, and each honeybee was observed for 6 s; therefore, the observation lasted up to 120 s for each cage with 20 bees.

### 2.6. Statistical Analysis

The GraphPad PRISM program (version 9.0, GraphPad program Inc., La Jolla, CA, USA) was used to perform statistical analysis. To ascertain or examine the events, the Kaplan–Meier estimate was employed. The Logrank test was used to compare the survival of the various treatment groups after the study was completed.

## 3. Results

### 3.1. BPF Analysis

Using HPLC, the polyphenol content of the BPF powder was determined. The three main flavonoids detected in BPF were neoeriocitrin (370 ppm), naringin (520 ppm), and neohesperidin (310 ppm). Standard microbiological tests (ISO 4833-1:2013 [28], ISO 21527-1:2008 [28], ISO 16694-2:2001 [28], ISO 4832:2006 [28], UNI EN ISO 6579:2000 [28], UNI EN ISO 6888-2:2004 [28], UNI EN ISO 7218:2007 [28]) showed that mycotoxins and dangerous bacteria were not present in the final BPF. The tests were negative for *Escherichia coli*, Coliforms, *Salmonella* and *Staphylococcus aureus*, while streptococci were reported as <1000 CFU/g in the aerobic plate count and yeasts/moulds <100 CFU/g in the count.

### 3.2. Survival Probability

Figure 1 represents the survival probability of the honeybees (*n* = 20 honeybees per cage) under treatment.

The survival of honeybees given the BPF doses did not differ statistically significantly (*p* > 0.05) from that of the control group (CTRL). BPF therefore was not found to be toxic at the three concentrations tested. Honeybees treated with FLU at both the 50 mg/L and the higher concentration of 100 mg/L showed a statistically significant (*p* < 0.001) lower probability of survival than the control group. The difference in survival probability was also statistically significant (*p* < 0.001) when comparing the two toxic doses of FLU. Indeed, as noted in Figure 1, in the group of honeybees receiving the toxic dose of 50 mg/L, 54% of the subjects managed to survive until the third day of treatment, while the bees given the 100 mg/L dose all died within two days of the initial administration. When honeybees were treated with both Flu and BPF (at different concentrations, 1, 2, 5 mg/kg), improved survival indices were noted in all groups tested. In particular, the three doses of BPF added to FLU 50 resulted in a statistically significant improvement with a *p* value of less than 0.05. The probability of survival improved further when comparing the results of the BPF-added groups to those in which the most toxic doses of FLU were administered (*p* value of less than 0.001). The survival and mortality rates of the experimental groups are summarized in Table 2 below.

### 3.3. Abnormal Behavior Pattern

Figure 2 represents the trend of abnormal behavior recorded by the treated honeybees during the experimentation.

The group whose diet was supplemented with BPF did not present statistically significant behavioral anomalies compared to the control group, while FLU at a dose of 50 mg/L caused abnormal behavior in a statistically significant manner (*p* < 0.001) compared to the control group. At the maximum tested dose of FLU (100 mg/L), all the bees showed abnormal behaviors; the statistical difference was significant (*p* < 0.01) compared with both the FLU 50 group and the control group (*p* < 0.001). When the toxic doses of FLU were combined with the three doses of BPF, behavioral improvements were noted. The three doses of BPF when associated with FLU 50 determined a statistically significant difference compared to the group treated with FLU 50 alone, a statistical difference which was revealed with a *p* value <0.01. When the three doses of BPF were associated with FLU 100, the difference was statistically significant at a *p* < 0.001 compared to the group administered FLU 100 alone.

### 3.4. Consumption of Treatments

Figure 3 represents the average consumption of the treatments administered for each experimental group.

The mean daily consumption of solution over the three days per bee in the control group was 0.06 mL. The BPF groups consumed on average as much as the control (*p* > 0.05). The FLU 50 group, however, consumed on average a statistically lower quantity than the control group (*p* < 0.01). The FLU 100 mg/L group consumed a statistically significant lower amount of solution compared to FLU 50 mg/L (*p* < 0.001). The FLU 50 mg/L group in combination with BPF consumed more solution than the FLU 50 mg/L group (*p* < 0.01) at all three concentrations. The FLU 100 and BPF association group consumed (*p* < 0.001) more than FLU 100 mg/L (*p* < 0.01).

## 4. Discussion

Pollination has an essential ecological function that helps to preserve the biodiversity of plants and animals [29]. Environmental, animal, and plant health are all strongly correlated with human health. This convergence explains why there should be alarm over the global entomofauna’s catastrophic fall in recent years [30]. Pesticides provide an undeniable risk to insects [31]. Researchers and agri-food policy makers have united to advocate for the use of more sustainable approaches for pest control in agriculture, such as integrated pest management (IPM) and biological control [32]. However, a long period of time is needed to consolidate this transition to new cultivation systems. Alongside those already on the market, new pesticides continue to be authorized after a risk assessment analysis. The purpose of the risk assessment of active substances is to ascertain whether these products do not degrade groundwater quality and whether they may have direct or indirect (e.g., through drinking water, food or feed) adverse effects on human or animal health. In addition, the environmental risk assessment seeks to determine how these products might affect non-target creatures in addition to the intended targets. Currently, the risk assessment of pesticides for honeybees in Europe ignores sub-lethal and/or chronic effects in favor of fatal ones [13]. It is still necessary to gain further insight into the effects that chemical exposure has on honeybees. Generally speaking, FLU is considered “bee safe” [11]. However, as reported by Tosi et al., 2021, FLU impaired bee survival and behavior at field-realistic doses (down to 11 ng/bee/day, corresponding to 400 µg/kg) that were up to 101-fold lower than those reported by risk assessments (1110 ng/bee/day), despite an absence of time-reinforced toxicity [13]. In such conditions, the discovery of products that may aid the body’s response to intoxication is an interesting solution to explore. Phenolic compounds play a major role in the antioxidant activity of many products [33,34]. According to a recent study by Hybl et al. in 2021, adding phenolic acids and flavonoids to the diet of bees intoxicated with thiacloprid extended their lifespan. This is likely because the increased expression level of genes encoding the cytochrome P450 enzyme increases the bees’ ability to detoxify [35]. The phenolic acid p-coumaric acid and the flavonoid quercetin have been shown in a 2017 study by Liao et al. to be able to upregulate detoxification enzymes in adult honeybees; their inclusion or lack in the diet may therefore have an impact on the toxicity of ingested pesticides [36]. The effects of dietary quercetin on the concentrations of three pesticides in honeybees—tebuconazole (fungicide), imidacloprid (insecticide), and tau-fluvalinate (insecticide and acaricide)—were also reported to be positive by Ardalani et al. in 2021 [37]. Bees’ detoxification system has been demonstrated to be regulated by quercetin. The researchers specifically demonstrated that quercetin consumption decreased the amount of imidacloprid present in honeybees [37]. Similar results were obtained by Liao et al. in 2020, who showed that groups of honeybees previously intoxicated with propiconazole and chlorantraniliprole had a much higher survival rate when their diet was supplied with phytochemicals [38]. Regardless of whether the experiments were short-term or long-term, the survival rate of the bees was positively influenced by the supplementation of natural bioactive compounds. Numerous studies have documented the potent bioactivities of citrus fruits, including their antibacterial, anti-inflammatory, and antioxidant properties [39,40,41]. Flavonoids make up more than 40% of the examined BPF; the other 60% is composed of other compounds, fatty acids, carbs, pectins, and maltodextrins. We have already demonstrated in a previous study that the incorporation of these flavonoids into the diet of honeybees is beneficial. The BPF specifically protected against deltamethrin intoxication [22]. In this study, two doses of FLU were tested, one dose that caused sub-lethal effects (50 mg/kg) and which allowed us to obtain behavioral anomalies and a dose that more easily led to mortality (100 mg/kg). Three doses of the BPF were then tested. In our previous study, only the 1 mg/kg dose was tested [22]. In this study, we verified that even the highest doses of 2 and 5 mg/kg are not toxic. However, increasing the dose did not produce an improvement in survival rates compared to the less concentrated dose. In any case, the BPF has proven to have great value. Its protective activity probably also extends to intoxication induced by other pesticides. The results of this experimental study show that honeybees intoxicated by FLU respond positively to the effects of intoxication. The treated subjects exhibited a lower percentage of abnormal behaviors when the BPF is associated with the diet. Furthermore, by virtue of the protection offered by the constituents of the BPF, the honeybees that ingest this “phytotherapeutic” managed to survive longer and did not suffer the inevitable mortality that occurred in the experimental groups in which FLU was administered alone. The BPF, at the concentrations tested, could therefore be administered to honeybees as an adjuvant in the response against the toxic effects of this pesticide. However, these are pilot studies and further experimentations are necessary before placing BPF-based preparations on the market for field administration.

## 5. Conclusions

The protection of honeybees is an important commitment to be pursued in order to preserve the integrity of terrestrial ecosystems. The harmful effects of pesticides result in extensive losses of honeybees to colonies, which then reverberate in the reduced efficiency of pollination services. The harm of pesticides must therefore be mitigated. This experimental study suggests that the implementation of biologically active substances can help support the organism’s response to the damage caused by poisoning. This can be the starting point for field studies aimed at determining the most appropriate doses to be administered to colonies.

## Figures and Tables

**Figure 1 animals-14-00608-f001:**
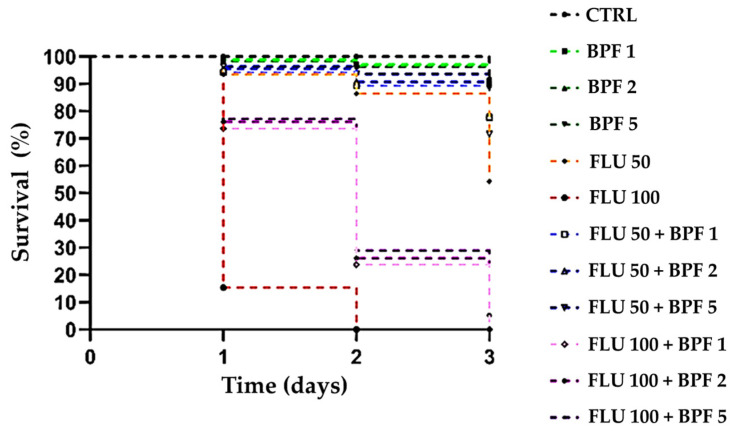
Survival graph. CTRL: control; BPF: bergamot polyphenolic fraction; FLU: flupyradifurone. BPF 1 vs. CTRL: *p* value > 0.05; BPF 2 vs. CTRL: *p* value > 0.05; BPF 5 vs. CTRL: *p* value > 0.05; FLU 50 vs. CTRL: *p* value < 0.001; FLU 100 vs. CTRL: *p* value < 0.001; FLU 100 vs. FLU 50: *p* value < 0.001; FLU 50 + BPF 1 vs. FLU 50: *p* value < 0.05; FLU 50 + BPF 2 vs. FLU 50: *p* value < 0.05; FLU 50 + BPF 5 vs. FLU 50: *p* value < 0.05; FLU 100 + BPF 1 vs. FLU 100: *p* value < 0.001; FLU 100 + BPF 2 vs. FLU 100: *p* value < 0.001; FLU 100 + BPF 5 vs. FLU 100: *p* value < 0.001.

**Figure 2 animals-14-00608-f002:**
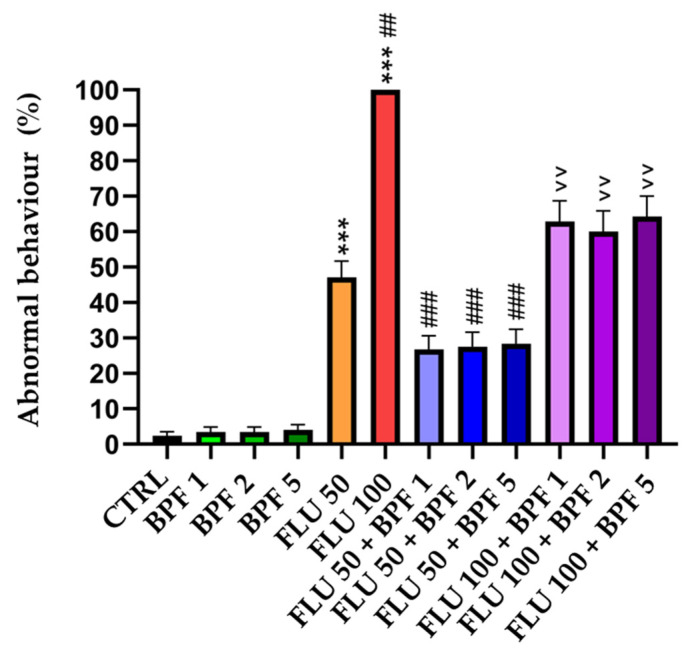
Abnormal behavior. CTRL: control; BPF: bergamot polyphenolic fraction; FLU: flupyradifurone. *** *p* value < 0.001 vs. CTRL; ## *p* value < 0.01 vs. FLU 50; ### *p* value < 0.01 vs. FLU 50; ^^ *p* value < 0.01 vs. FLU 100.

**Figure 3 animals-14-00608-f003:**
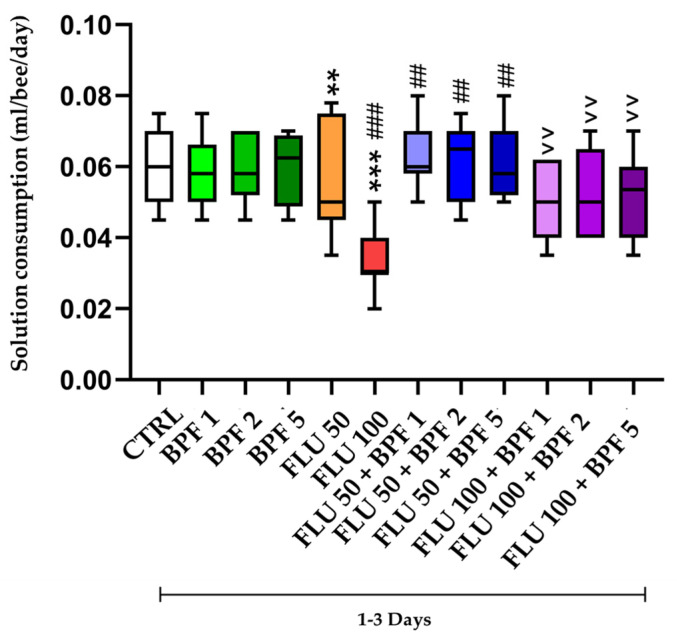
Treatment consumption. CTRL: control; BPF: bergamot polyphenolic fraction; FLU: flupyradifurone. ** *p* value < 0.01 vs. CTRL; *** *p* value < 0.001 vs. CTRL; ## *p* value < 0.01 vs. FLU 50; ### *p* value < 0.001 vs. FLU 50; ^^ *p* value < 0.001 vs. FLU 100.

**Table 1 animals-14-00608-t001:** Experimental groups.

Group	Flupyradifurone (FLU)	Bergamot Polyphenolic Fraction (BPF)	Sucrose Solution
CTRL	-	-	50% *w*/*v*
FLU 50	50 mg/L	-	50% *w*/*v*
FLU 100	100 mg/L	-	50% *w*/*v*
BPF 1	-	1 mg/kg	50% *w*/*v*
BPF 2	-	2 mg/kg	50% *w*/*v*
BPF 5	-	5 mg/kg	50% *w*/*v*
FLU 50 + BPF 1	50 mg/L	1 mg/kg	50% *w*/*v*
FLU50 + BPF 2	50 mg/L	2 mg/kg	50% *w*/*v*
FLU 50 + BPF 5	50 mg/L	5 mg/kg	50% *w*/*v*
FLU 100 + BPF 1	100 mg/L	1 mg/kg	50% *w*/*v*
FLU 100 + BPF 2	100 mg/L	2 mg/kg	50% *w*/*v*
FLU 100 +BPF 5	100 mg/L	5 mg/kg	50% *w*/*v*

**Table 2 animals-14-00608-t002:** Survival, mortality and standard deviation (±) of the experimental groups.

Treatment	Day	Survival (%)	Mortality (%)
CTRL	1	100 ± 0	0 ± 0
	2	100 ± 0	0 ± 0
	3	91.6 ± 0.03	8.4 ± 0.03
BPF 1	1	98.8 ± 0.03	1.2 ± 0.03
	2	97.1 ± 0.03	2.9 ± 0.03
	3	90.1 ± 0.03	9.9 ± 0.03
BPF 2	1	98.2 ± 0.05	1.8 ± 0.05
	2	96.5 ± 0.03	3.5 ± 0.03
	3	89.6 ± 0.02	10.4 ± 0.02
BPF 5	1	98.4 ± 0	1.6 ± 0
	2	96.2 ± 0.41	3.8 ± 0.41
	3	88.1 ± 0.03	12.9 ± 0.03
FLU 50	1	93.4 ± 0.13	6.6 ± 0.13
	2	86.4 ± 0.06	13.6 ± 0.06
	3	54.2 ± 0.37	45.8 ± 0.37
FLU 100	1	15.3 ± 0.03	84.7 ± 0.03
	2	0 ± 0	100 ± 0
	3	/	/
FLU 50 + BPF 1	1	94.2 ± 0.05	5.8 ± 0.05
	2	89.4 ± 0.08	10.6 ± 0.08
	3	77.7 ± 0.09	22.3 ± 0.09
FLU 50 + BPF 2	1	95.5 ± 0.06	4.5 ± 0.06
	2	90.6 ± 0.08	9.4 ± 0.08
	3	79.1 ± 0.15	20.9 ± 0.15
FLU 50 + BPF 5	1	96.3 ± 0.41	3.7 ± 0.41
	2	93.6 ± 0.13	6.4 ± 0.13
	3	82.0 ± 0.10	18 ± 0.10
FLU 100 + BPF 1	1	73.6 ± 0.18	26.4 ± 0.18
	2	26.4 ± 0.13	73.6 ± 0.13
	3	0 ± 0	100 ± 0
FLU 100 + BPF 2	1	76.0 ± 0.15	24 ± 0.15
	2	26.1 ± 0.16	73.9 ± 0.16
	3	5.2 ± 0.53	94.8 ± 0.53
FLU 100 + BPF 5	1	77.1 ± 0.18	22.9 ± 0.18
	2	28.9 ± 0.21	71.1 ± 0.21
	3	0 ± 0	100 ± 0

## Data Availability

The data are kept at the University of Magna Græcia of Catanzaro and are available upon request.

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
