# Peer review of "Bergamot Polyphenolic Fraction for the Control of Flupyradifurone-Induced Poisoning in Honeybees"

_animals, 2024, doi:10.3390/ani14040608_

Round 1

Reviewer 1 Report

Comments and Suggestions for Authors

This is a quite interesting and promising study concerning testing polyphenolic fractions for control of poisoning honeybees.

The main flaw in the study is lack of repetitions. I would expect at least three groups of bees for each treatment. Then we would have average survival rate, average mortality rate, standard deviation, and in such case statistical analysis would be very helpful and clear. The data which you presented in table 2 do not show that you did any repetitions of your experiment.

Authors should add explanation about why such doses of FLU where applied and how it refers to field-realistic doses. I suppose it is easy  to calculate but it would shed a new light on the results which they obtained

Besides, you should give more details concerning how the composition of flavonoid profile was analyzed.

Change Kg to kg

Line 109

It is: …range from 93-66%, it should be: …range from 66 to 99%.

There is general rule that we don't repeat data in figures and tables, from my perspective first two figures are not necessary, table 2 is enough

You should add statistical symbols from the figure 3 to figure caption or add legend like in figure 4

Author Response

REVIEWER 1

This is a quite interesting and promising study concerning testing polyphenolic fractions for control of poisoning honeybees.

Response: we thank you for your appreciation of our manuscript and for your revision work, which helps us to profoundly improve the quality of the document. We have made changes following your advice. The changes we have made are highlighted in the new document.

The main flaw in the study is lack of repetitions. I would expect at least three groups of bees for each treatment. Then we would have average survival rate, average mortality rate, standard deviation, and in such case statistical analysis would be very helpful and clear. The data which you presented in table 2 do not show that you did any repetitions of your experiment.

R: the information about the experimental replicates is specified on lines 153. There were 10 experimental replicates for each group. To clarify the concept, standard deviations have been included in table 2. This latter information was missing in the document initially submitted.

Authors should add explanation about why such doses of FLU where applied and how it refers to field-realistic doses. I suppose it is easy to calculate but it would shed a new light on the results which they obtained

R: thank you for this advice, which helps us improve the overall quality of the manuscript. More details have been added in the manuscript at line 150.

Besides, you should give more details concerning how the composition of flavonoid profile was analyzed.

R: thanks for this important advice. To provide more details, a specific paragraph (2.3. BPF analysis) has been added

Change Kg to kg

R: amended

Line 109

It is: …range from 93-66%, it should be: …range from 66 to 99%.

R: amended

There is general rule that we don't repeat data in figures and tables, from my perspective first two figures are not necessary, table 2 is enough

R: with your consent we would prefer to keep at least the first graph. This image has a visual impact that best gives an idea of the improvements we have seen following BPF treatment. We have, on the other hand, removed the second graph which showed the same data from a different point of view, but was redundant.

You should add statistical symbols from the figure 3 to figure caption or add legend like in figure 4

R: the legend has been modified

Reviewer 2 Report

Comments and Suggestions for Authors

Dear authors,

In your manuscript you described ‘’ Bergamot polyphenolic fraction for the control of flupyradifuron-induced poisoning in honeybees’’. In my opinion, your article needs some correction. Here are my comments and suggestions:

Please separate the text at the introduction because is only one paragraph

Line 136: How many groups did you create?

Line 143: it is better to write one day before the beginning of the experiment

Line 151: What do you mean with 72? The experiment took place only for 3 days? Please make clear the duration of the experiment in the begging of the experimental part.

Lines 175-176: Please be more specific explaining shortly the analysis conditions, if you use calibration curves etc  

Lines 203-204: Be more specific

Line 219: Did you use Orbitrap spectrometer or HPLC? If you use HPLC explain the procedure in material and methods (Conditions, calibration curves, columns etc.)

Figures: In my opinion you should not use p values in your figures because the legends are getting big

Table 2: make it more compressed

Discussion: make smaller paragraphs because is easier to the readers

Author Response

REVIEWER 2

In your manuscript you described ‘’ Bergamot polyphenolic fraction for the control of flupyradifuron-induced poisoning in honeybees’’. In my opinion, your article needs some correction. Here are my comments and suggestions:

Response: thank you for your important revision work. We have made the changes that you will find highlighted in the new attached document.

Please separate the text at the introduction because is only one paragraph

R: modified as suggested

Line 136: How many groups did you create?

R: twelve groups (detailed in table 1) were created and 10 test replicates were performed for each experimental group. The information about the experimental replicates is specified on lines 153. To clarify the concept, standard deviations have been included in table 2. This latter information was missing in the document initially submitted.

Line 143: it is better to write one day before the beginning of the experiment

R: amended as suggested

Line 151: What do you mean with 72? The experiment took place only for 3 days? Please make clear the duration of the experiment in the begging of the experimental part.

R: this information has been clarified as you can verify at line 156

Lines 175-176: Please be more specific explaining shortly the analysis conditions, if you use calibration curves etc  

R: thanks for this important advice. To provide more details, a specific paragraph (2.3. BPF analysis) has been added

Lines 203-204: Be more specific

R: thank you for this advice. To clarify the concept, the bibliographical reference has been added and the sentence was modified.

Line 219: Did you use Orbitrap spectrometer or HPLC? If you use HPLC explain the procedure in material and methods (Conditions, calibration curves, columns etc.)

R: thanks for this important advice. To provide more details, a specific paragraph (2.3. BPF analysis) has been added

Figures: In my opinion you should not use p values in your figures because the legends are getting big

R: the legends of figures 2 and 3 have been modified to make them shorter. For Figure 1, with your permission, we would prefer to keep the p values because they explain the graph.

Table 2: make it more compressed

R: the table has been modified and the dimensions have been adapted to the content.

Discussion: make smaller paragraphs because is easier to the readers

R: thanks for this advice. The longer sentences have been shortened to make the concepts explained better understandable.

Round 2

Reviewer 1 Report

Comments and Suggestions for Authors

Authors did some corrections which generally are satisfying but still I believe that you can better present your results e.g., figure 1 and table 1 should be changed into column figures where the statistical analysis would be easier to presented. Now, it is a little bit messy and complicated for the reader to see clearly your results.

Author Response

Dear Reviewer,

We thank you for the advice, but the choice of the type of graphical representation of the results was not random but constrained by the type of analysis performed. The Kaplan-Meier estimator, also known as the product limit estimator, is a non-parametric statistic used to estimate the survival function from lifetime data. In fact, when creating a Kaplen-Meier survival curve, the software automatically generates a type of graph that can only be scaled or stepped. The bar graph would allow us to represent only a cumulative mortality and not a mortality spread over time, i.e. over the three days of experimentation. You can verify that the Kaplen-Meier graph is the best for these types of studies by consulting the following manuscripts: https://doi.org/10.1897/02-578, https://doi.org/10.3390/insects12040357, https://doi.org/10.1111/1462-2920.12825, https://doi.org/10.1098/rspb.2019.0433, https://doi.org/10.1038/s42003-021-02336-2. Therefore, we ask your permission to keep the current figure 1 and table 1.

Reviewer 2 Report

Comments and Suggestions for Authors

Thank you for your corrections 

Author Response

Dear reviewer,

thank you for the advice given and for the appreciation of the manuscript.